# Artificial Intelligence-Enabled Electrocardiography Detects B-Type Natriuretic Peptide and N-Terminal Pro-Brain Natriuretic Peptide

**DOI:** 10.3390/diagnostics13172723

**Published:** 2023-08-22

**Authors:** Pang-Yen Liu, Chin Lin, Chin-Sheng Lin, Wen-Hui Fang, Chia-Cheng Lee, Chih-Hung Wang, Dung-Jang Tsai

**Affiliations:** 1Division of Cardiology, Department of Internal Medicine, Tri-Service General Hospital, National Defense Medical Center, Taipei 114, Taiwan; liupydr@gmail.com (P.-Y.L.); littlelincs@gmail.com (C.-S.L.); 2Medical Technology Education Center, School of Medicine, National Defense Medical Center, Taipei 114, Taiwan; xup6fup0629@gmail.com; 3School of Public Health, National Defense Medical Center, Taipei 114, Taiwan; 4Artificial Intelligence of Things Center, Tri-Service General Hospital, National Defense Medical Center, Taipei 114, Taiwan; rumaf.fang@gmail.com; 5Department of Family and Community Medicine, Department of Internal Medicine, Tri-Service General Hospital, National Defense Medical Center, Taipei 114, Taiwan; 6Medical Informatics Office, Tri-Service General Hospital, National Defense Medical Center, Taipei 114, Taiwan; lcgnet@gmail.com; 7Division of Colorectal Surgery, Department of Surgery, Tri-Service General Hospital, National Defense Medical Center, Taipei 114, Taiwan; 8Department of Otolaryngology–Head and Neck Surgery, Tri-Service General Hospital, National Defense Medical Center, Taipei 114, Taiwan; chw@ms3.hinet.net; 9Graduate Institute of Medical Sciences, National Defense Medical Center, Taipei 114, Taiwan; 10Department of Statistics and Information Science, Fu Jen Catholic University, New Taipei City 242, Taiwan

**Keywords:** artificial intelligence, electrocardiogram, deep learning, B-type natriuretic peptide, N-terminal pro-brain natriuretic peptide

## Abstract

**BACKGROUND:** The B-type natriuretic peptide (BNP) and N-terminal pro-brain natriuretic peptide (pBNP) are predictors of cardiovascular morbidity and mortality. Since the artificial intelligence (AI)-enabled electrocardiogram (ECG) system is widely used in the management of many cardiovascular diseases (CVDs), patients requiring intensive monitoring may benefit from an AI-ECG with BNP/pBNP predictions. This study aimed to develop an AI-ECG to predict BNP/pBNP and compare their values for future mortality. **METHODS:** The development, tuning, internal validation, and external validation sets included 47,709, 16,249, 4001, and 6042 ECGs, respectively. Deep learning models (DLMs) were trained using a development set for estimating ECG-based BNP/pBNP (ECG-BNP/ECG-pBNP), and the tuning set was used to guide the training process. The ECGs in internal and external validation sets belonging to nonrepeating patients were used to validate the DLMs. We also followed-up all-cause mortality to explore the prognostic value. **RESULTS:** The DLMs accurately distinguished mild (≥500 pg/mL) and severe (≥1000 pg/mL) an abnormal BNP/pBNP with AUCs of ≥0.85 in the internal and external validation sets, which provided sensitivities of 68.0–85.0% and specificities of 77.9–86.2%. In continuous predictions, the Pearson correlation coefficient between ECG-BNP and ECG-pBNP was 0.93, and they were both associated with similar ECG features, such as the T wave axis and correct QT interval. ECG-pBNP provided a higher all-cause mortality predictive value than ECG-BNP. **CONCLUSIONS:** The AI-ECG can accurately estimate BNP/pBNP and may be useful for monitoring the risk of CVDs. Moreover, ECG-pBNP may be a better indicator to manage the risk of future mortality.

## 1. Introduction

Globally, more than 26 million individuals are afflicted by heart failure (HF) and cardiac dysfunction, constituting significant challenges to public health on a global scale. Given their elevated morbidity and mortality rates, diagnosing HF and cardiac dysfunction holds paramount importance in both clinical and forensic medicine [1,2]. The incidence of HF is on the rise, attributed to a growing elderly population and an increase in patients with chronic HF who have survived acute episodes due to advancements in therapeutic interventions [3,4]. As a result, HF holds immense significance from both medical and socioeconomic perspectives [5].

Echocardiography serves as the benchmark for heart failure (HF) diagnosis, being the most valuable method and a cornerstone for guiding diagnostic and therapeutic decisions. Nevertheless, echocardiography’s drawbacks include its high costs and limited effectiveness in patients with obesity or concurrent chronic lung diseases accompanied by respiratory distress. As a result, biochemical markers have evolved into essential clinical tools, simplifying and enhancing the accuracy of HF diagnosis and prognosis by delineating its occurrence and stage. The B-type natriuretic peptide (BNP) and N-terminal pro-BNP (pBNP) have become routine laboratory tests, extensively studied for their clinical utility. They have demonstrated effectiveness in distinguishing severe acute respiratory distress syndrome and in diagnosing and assessing HF [6,7]. However, it is worth noting that both the traditional and newly reported biochemical markers have certain limitations as diagnostic and prognostic tools, leading to a constrained overall applicability.

It was discovered in 1988 that a peptide from the porcine brain exhibited a natriuretic effect, which was named BNP [8]. The 32-amino acid C-terminal fragment of BNP is synthesized from pBNP, which is produced by ventricular cardiomyocytes in response to fluid overload in HF [9,10]. Subsequent studies showed that the level of BNP in plasma was increased in patients with HF and increased with the severity of HF [10,11]. Larger cohort studies confirmed that plasma BNP was significantly increased in patients with systolic dysfunction and identified plasma BNP as an accurate marker of HF due to the strong negative predictive value of a low BNP result [12,13,14]. pBNP is an inert 76-amino acid N-terminal fragment, and the other circulating fragment is rapidly cleaved from pBNP secreted by cardiomyocytes in response to the HF increased wall stretch [15,16]. Initial studies have revealed the value of pBNP in the emergency diagnosis of acute destabilized HF presenting acute dyspnea [17,18,19]. In chronic and stable HF, pBNP has been shown to be strongly correlated with BNP as a good predictor of the outcome and superior to BNP for predicting mortality and morbidity [20]. Although both fragments, BNP and pBNP, are routinely used to assist diagnosis and predict outcomes due to their good availability, pBNP has advantages in its longer half-life and a greater range of values [16]. However, the elevation of plasma BNP or pBNP in other conditions, such as atrial fibrillation, chronic lung disease, and sleep apnea, makes echocardiography a valuable tool for the diagnosis of HF, even though echocardiography has low availability and relatively high costs [21,22].

The resting 12-lead electrocardiogram (ECG) is another potential tool to aid in the diagnosis of HF [23,24]. Similar to the high negative predictive value of BNP/pBNP, normal ECG results have been used to exclude left ventricular systolic dysfunction in patients with suspected HF [25]. The AI-enabled ECG has been studied in an acute care setting for detecting HF and, to our knowledge, has been less evaluated for its correlation with BNP/pBNP and a long-term clinical outcome [26]. We, therefore, sought to evaluate the correlation between the 12-lead resting ECG and the plasma concentration of BNP/pBNP with the predictive value of the outcome in HF.

## 2. Methods

### 2.1. Data Source and Population

This retrospective study was ethically approved by the institutional review board of Tri-Service General Hospital, Taipei, Taiwan (IRB NO. C202105049). We reviewed the patients who visited our hospital during January 2010 and September 2021 from the electronic medical records (EMRs). Patients who simultaneously received ECG examinations and BNP/pBNP tests within 24 h were included in this study. The BNP and pBNP tests were not simultaneously conducted in our hospital, and physicians made the decision to use only one of the tests for assessing the cardiac risk of patients. Therefore, the records with a BNP test did not have pBNP information in this study, and vice versa.

Figure 1 shows the algorithm used to generate each dataset in this study. A total of 29,822 patients in an academic medical center (Hospital A, Neihu General Hospital at Neihu District) met the above criteria, and 20,789 patients after January 2017 were assigned to a development set to train DLMs. Of the 47,709 ECGs, 19,611 had corresponding BNP annotations, and 28,098 had corresponding pBNP annotations. The 5032 patients during January 2016 and December 2016 provided 16,249 ECGs for guiding the training process as an independent tuning set, comprised of 10,695 ECGs with BNP annotations and the remaining 5554 ECGs with pBNP annotations. The internal validation set used 4001 patients before December 2015 to maximize the follow-up period, and we only used the first record of each patient to eliminate data dependency. There were 3090 and 911 records in the BNP and pBNP subsets, respectively. A community hospital (Hospital B, Tingzhou Branch Hospital at Zhongzheng District) provided 6042 patients with the first ECG as an external validation set. The numbers of patients in the BNP and pBNP subsets were 3966 and 2076, respectively.

### 2.2. Data Collection

All ECGs in this study were measured by the standard 12-lead ECG Philips machine (PH080A) with a 500 Hz sampling frequency for 10 s and stored in a digital format. Due to the “black box” property of DLMs, 8 quantitative ECG measures and 31 diagnostic pattern classes were also collected from the structured findings statements based on the key phrases that are standard within the Philips system. These features were used to train an extreme gradient boosting (XGB) model, and the DLM was trained via raw ECG traces. Pattern classes included the abnormal T wave, atrial fibrillation, atrial flutter, atrial premature complex, complete AV block, complete left bundle branch block, complete right bundle branch block, first degree AV block, incomplete left bundle branch block, incomplete right bundle branch block, ischemia/infarction, junctional rhythm, left anterior fascicular block, left atrial enlargement, left axis deviation, left posterior fascicular block, left ventricular hypertrophy, low QRS voltage, pacemaker rhythm, prolonged QT interval, right atrial enlargement, right ventricular hypertrophy, second degree AV block, sinus bradycardia, sinus pause, sinus rhythm, sinus tachycardia, supraventricular tachycardia, ventricular premature complex, ventricular tachycardia, and Wolff–Parkinson–White syndrome. The detailed items have been described in previous works [27,28]. Blood BNP and pBNP were based on central laboratory methods. Since there are no universal cutoff points for BNP and pBNP, we divided the data into three categories based on BNP/pBNP by the same values: normal BNP/pBNP (<500 pg/mL), mild abnormal BNP/pBNP (500–999 pg/mL), and severe abnormal BNP/pBNP (≥1000 pg/mL).

We additionally collected demographic information, including sex, age, and body mass index, and disease histories, including diabetes mellitus (DM), hypertension (HTN), hyperlipidemia (HLP), chronic kidney disease (CKD), acute myocardial infarction (AMI), stroke (STK), coronary artery disease (CAD), heart failure (HF), atrial fibrillation (Afib), and chronic obstructive pulmonary disease (COPD), based on the corresponding International Classification of Diseases, Ninth Revision and Tenth Revision (ICD-9 and ICD-10, respectively) from EMRs. The detailed look-up tables for ICD codes and diseases have been described previously [29,30,31,32,33].

All-cause mortality events were also collected in this study. The survival time was calculated with reference to the date of the ECG test. Patient status (dead/alive) was captured through the EMRs. Data for alive visits were censored at the patient’s last known hospital alive encounter to limit bias from incomplete records. The end of follow-up in this study was 31 September 2021. The median follow-up periods were 25.2 months (interquartile range [IQR]: 6.3–46.5 months) and 15.3 months (IQR: 2.0–38.2 months) in the internal and external validation sets, respectively, with maximum follow-up times of 123.8 months and 120.7 months. The number of mortality events was 463 (3.93 per 1000 person-years) and 666 (4.59 per 1000 person-years) in the internal and external validation sets, respectively.

### 2.3. The Implementation of the Deep Learning Model

We directly trained DLMs using 12-lead ECG trace signals to estimate BNP and pBNP. We used the architecture of ECG12Net, which has been previously described in detail [34]. Appendix A illustrates the architecture of our Deep Learning Model (DLM). Each electrocardiogram (ECG) was captured in the standard 12-lead format, comprising sequences of 5000 data points. These sequences were used to create a matrix of a size of 5000 × 12. The input format of this architecture was a 4096 × 12 matrix. During the training process, we randomly selected sequences of a length of 4096 as the input. In the inference stage, two overlapping sequences of 4096 were utilized from both the beginning and the end to generate predictions, which were then averaged to produce the final prediction.

We defined a “residual module” as a neural combination with a constant factor *k*. This module included the following components: (1) a 1 × 1 convolutional layer with *k*/4 filters for dimension reduction, (2) a batch normalization layer for normalization, (3) a rectified linear unit (ReLU) layer for introducing non-linearity, (4) a 3 × 1 convolutional layer with *k*/4 filters to extract features, (5) a batch normalization layer for normalization, (6) a ReLU layer for non-linearity, (7) a 3 × 1 convolutional layer with 4K filters to further extract features, (8) a 1 × 1 convolutional layer with *k* filters to restore feature shape, (9) a batch normalization layer for normalization, (10) a ReLU layer for non-linearity, and (11) a squeeze-and-excitation (SE) module for feature weighting. The SE module comprised: (1) an average global pooling layer, (2) a fully-connected layer with *k*/*r* neurons, and (3) another fully-connected layer with *k* neurons. The constant r remained fixed at 8 across all experiments. The residual module concluded with a shortcut connection, creating direct connections between each layer and all subsequent layers.

When there is a change in the size of feature maps, concatenating the residual module becomes infeasible. As a solution, our architecture employs a “pool module” to facilitate the concatenation of each residual module during down-sampling. This module consists of similar concatenated layers as the residual modules, but the stride of the 3 × 1 convolution layer is altered to 2 × 1. Down-sampling is achieved through an average pooling layer with a 2 × 1 kernel size and stride. These components are integrated using the concatenation function.

The input data first pass through a batch normalization layer, followed by an 11 × 1 convolution layer with a 2 × 1 stride and 16 filters, another batch normalization layer, a ReLU layer, and a pool module. Subsequently, the data traverse a series of residual modules and pool modules, yielding a 32 × 12 × 1024 array. This is followed by a global pooling layer and the last residual module. The array is then split into 12 lead-specific feature maps, each containing 1024 features. These feature maps undergo processing via a fully-connected layer with one neuron to generate lead-specific predictions.

To enhance the interpretive capacity of the DLM, an attention mechanism based on a hierarchical attention network is employed to combine these blocks. The attention module involves a fully connected layer with 8 neurons, followed by a batch normalization layer, a ReLU layer, and another fully-connected layer with one neuron that generates weights for each lead. Attention scores are computed for each ECG lead, and then integrated and standardized through the final linear output layer. These standardized attention scores are used to weight the outputs of the 12 ECG leads via simple multiplication. The weighted outputs are summed and processed through a prediction module to yield the final prediction value.

During training, these DLMs use a batch size of 32 and an initial learning rate of 0.001, with an Adam optimizer employing standard parameters (β1 = 0.9 and β2 = 0.999). The learning rate undergoes decay by a factor of 10 whenever the validation cohort’s loss plateaus after an epoch. To curb overfitting, we employ early stopping by saving the network after each epoch and selecting the DLMs with the lowest loss on the validation cohort. In this study, the sole regularization method for preventing overfitting is L2 regularization with a coefficient of 10^−4^. We trained two types of DLMs, binary output and continuous output, to conduct the corresponding tasks. In continuous predictions, the categorical encoding technology and the training details used were according to previous studies [23,35,36], and we converted the values of BNP and pBNP to a log scale due to their significant right-skewed distributions. The value range of BNP was limited from 10 to 5000 pg/mL, and the value range of pBNP was limited from 10 to 35,000 pg/mL. The original out-of-range values were revised to the boundary value for all analyses.

### 2.4. Statistical Analysis

The presentation of patient characteristics included means and standard deviations for continuous variables, along with numbers and percentages for categorical variables. The statistical analysis was conducted using R version 3.4.4, and the implementation of our Deep Learning Models (DLMs) was done using the MXNet package version 1.3.0. To evaluate the performance of the DLMs in identifying mild and severe abnormal BNP/pBNP, we employed receiver operating characteristic (ROC) curves and calculated areas under the curve (AUCs). The sensitivity, specificity, positive predictive value, and negative predictive value were determined based on the maximum Youden’s index from the tuning set. The selected operating point was consistently used for the analysis in both internal and external validation sets. For the continuous analysis, scatter plots and Pearson correlation coefficients (r) were used to compare the predictions from the electrocardiogram (ECG) with actual BNP/pBNP values. The importance analysis of ECG features was initiated with an eXtreme gradient boosting (XGB) model for predicting ECG-BNP/ECG-pBNP, resulting in the presentation of feature importance rankings. The relationship between ECG features and ECG-BNP/ECG-pBNP was visually represented using a bar chart with quartiles. In the context of the mortality analysis, a Kaplan–Meier (KM) curve was generated to assess the prognostic contribution of ECG-BNP/ECG-pBNP. Additionally, Cox proportional hazard models were fitted to calculate grouping hazard ratios (HRs), along with corresponding 95% confidence intervals (95% CI). The C-index was employed as a global indicator to compare the predictive performance between ECG-BNP and ECG-pBNP.

## 3. Results

Patient internal and external validation sets were distinguished as BNP and pBNP subsets. There were 8.8%/10.2% and 10.2%/11.8% patients with mild and severe abnormal BNP in the internal/external validation set, respectively, and there were 9.0%/26.8% and 10.1%/27.7% patients with mild and severe abnormal pBNP in the internal/external validation set, respectively. The mean age of these datasets ranged from 68.9 to 74.0 years, and the proportion of males ranged from 49.6% to 53.0%. The other detailed distributions are shown in Table 1.

Figure 2 shows the performance of DLMs with a binary output. Similarly high AUCs of ≥0.85 were presented in all analyses. The BNP model (AUC = 0.8831/0.8934) performed better than the pBNP model (AUC = 0.8566/0.8547) in the internal validation set, but the pBNP model (AUC = 0.8802/0.8787) performed better than the BNP model (AUC = 0.8571/0.8524) in the external validation set. The sensitivities ranged from 68.0–85.0%, and specificities ranged from 77.9–86.2%. Moreover, the DLMs did not perform significantly better in detecting severe abnormal BNP/pBNP compared to mild abnormalities. This explained that the DLMs may estimate the accurate continuous value of BNP/pBNP by linear prediction models.

Figure 3 shows the performance of DLMs with a linear output. The Pearson correlation coefficients between the actual BNP and ECG-BNP predicted by DLMs were 0.72 and 0.68 in the internal and external validation sets, respectively. Moreover, the Pearson correlation coefficients between the actual pBNP and ECG-pBNP predicted by DLMs were 0.69 and 0.73 in the internal and external validation sets, respectively. Since the DLMs with a binary output also performed similarly in detecting abnormal BNP and pBNP, the relationship between ECG and BNP/pBNP might be considered similar. We further analyzed the association between ECG-BNP and ECG-pBNP, and a high Pearson correlation coefficient of 0.93 was presented in both the internal and external validation sets. The above results emphasized a similarity in ECG-BNP and ECG-pBNP. Considering the use of BNP and pBNP in clinical practice, we may only need one of them to manage the risk of cardiovascular diseases.

Figure 4 demonstrates the relationship between known ECG features and ECG-BNP. Known ECG features explained 59.84% and 55.26% of the variation in ECG-BNP, which explained why some features could be extracted by DLMs. Our DLMs identified patients with a higher ECG-pBNP who were associated with a higher T wave axis and prolonged correct QT interval. Moreover, patients without a sinus rhythm were considered to have a higher ECG-BNP. Although the other known ECG features might just contribute less to ECG-BNP, we identified that patients with a prolonged QRS duration, increased heart rate, shortened PR interval, and lower RS wave axis were related to a higher ECG-BNP. Patients with left ventricular hypertrophy and atrial fibrillation were also considered to have a higher ECG-BNP. A similar analysis was also conducted for ECG-pBNP, as shown in Figure 5. The ECG-pBNP shared the same known ECG features with ECG-BNP. However, the most important difference between these two analyses was that the proportion of variation could be explained. Known ECG features only explained 52.40% and 52.28% of the variation in ECG-pBNP. This might imply that more unknown information was identified by ECG-pBNP.

Figure 6 shows the prognostic values of ECG-BNP and ECG-pBNP for mortality events. The incidence of all-cause mortality was 14.8% at 2 years and 26.9% at 8 years in the Q4 group of ECG-BNP in the internal validation set, which was significantly higher than that in the Q1 group (4.2% and 11.6%, respectively), with a sex-age adjusted HR of 2.76 (95% CI: 2.02–3.78). The external validation analysis showed similar results, and an obvious dose-effect was presented to emphasize the importance of cardiovascular disease management by ECG-BNP. However, ECG-pBNP also showed a prognostic value for mortality events. The HR of the Q4 group was 2.79 (95% CI: 2.05–3.80) compared to the Q1 group in the internal validation set in the ECG-pBNP analysis, and the result was also validated in the external analysis. Importantly, the superiority of ECG-pBNP with the sex and age adjustment was evident compared to ECG-BNP, showing higher C-indices of 0.655/0.665 compared to 0.637/0.640 in the internal/external validation sets, respectively. These results demonstrated that ECG-pBNP had a better prognostic value for future mortality than ECG-BNP.

## 4. Discussion

DLMs show the promising ability to capture BNP/pBNP information from raw ECG signals, and a strong and continuous relationship was shown. This study simultaneously used DLMs to estimate ECG-BNP and ECG-pBNP, and the association analysis showed similar concepts between them. However, there remained a small difference between ECG-BNP and ECG-pBNP. The latter captured more unknown ECG features than ECG-BNP, which might contribute a high prognostic value to mortality prediction in further follow-up analyses. In summary, the AI-enabled ECG system could accurately estimate ECG-pBNP and provide more information to manage future cardiovascular diseases.

To help diagnose acute HF, the ESC guidelines recommend that all patients with suspected acute HF have their plasma natriuretic peptide levels (BNP and pBNP) measured. BNP has a normal upper limit of 35 pg/mL in the nonacute setting, while pBNP has a normal upper limit of 300 pg/mL in the nonacute setting and 125 pg/mL in the acute setting [37]. In clinical practice, BNP levels can help clinicians distinguish between HF-related dyspnea and other causes. It is considered unlikely that HF is responsible for dyspnea if BNP exceeds 100 pg/mL. If the BNP level falls within the range of 100 to 500 pg/mL, diagnosing HF should rely on clinical judgment. When BNP exceeds 500 pg/mL, the possibility of HF or cardiac dysfunction arises, and prompt HF therapy is recommended [38]. The ICON study, focusing on NT-proBNP, indicates that utilizing age-specific pBNP cutoffs could enhance HF diagnosis. A universal pBNP cutoff of 300 pg/mL could serve to rule out acute HF, irrespective of age. There are three main criteria for diagnosing HF: those under 50 years of age with pBNP levels > 450 pg/mL, those between 50 and 75 years of age with pBNP levels > 900 pg/mL, and those over 75 years of age with pBNP levels > 1800 pg/mL [39]. The results of our study also showed good accuracy when severity judgments were made using a similar cutoff point. Even for pBNP, the accuracy of the external analysis was higher than that of the internal analysis.

There has been a widespread use of BNP and pBNP as clinical indicators of heart failure and cardiac dysfunction [40,41,42]. Previous retrospective research has shown that QRS duration alone is less sensitive than BNP to detect systolic heart failure as a screening test [43]. We extended the DLMs to include additional features such as the T wave axis, corrected QT interval, left ventricular hypertrophy, and atrial fibrillation to identify patient groups with high ECG-BNP/ECG-pBNP levels. Cardiovascular dysfunction and HF are diagnosed mainly with BNP and pBNP biomarkers. The severity of heart disease, treatment strategies, and prognosis are also determined by these factors [37,44]. Patients with dyspnea in the emergency room can be identified with systolic heart failure using an AI-enabled ECG, but long-term outcomes have not been evaluated by the AI-enabled ECG [26]. Our DLMs revealed superiorities of ECG-pBNP in predicting mortality risk 8 years beyond the acute care setting. This amazing ability of the AI-ECG to capture useful features to help diagnosis and prognosis prediction beyond human capabilities has also been demonstrated in previous studies [45,46,47,48,49]. Furthermore, the benefit of embedding the AI-ECG into a hospital information system as a passive notification system has been validated in clinical practice [32]. We considered that our AI-ECG may help physicians identify more patients with high BNP or pBNP to make a diagnosis in the early stage of acute HF.

The concentration of BNP is closely related to the incidence and severity of heart failure; Indeed, its magnitude escalates as the severity rises, a classification established by the New York Heart Association's functional assessment. Therefore, it can be used to predict a patient’s status and develop a treatment plan [50]. In the process of therapeutic decision-making, the predictive power of a diagnostic test plays a crucial role in determining the test’s diagnostic value. Based on this analysis, BNP is a significant predictor of heart failure and myocardial infarction mortality rates [51]. Heart failure can be predicted by this clinical outcome more accurately than by any other prognostic factor, and BNP is thought to be an important prognostic factor for future trials [6]. Similarly, ECG-BNP had good predictive power for future mortality risks in our study. It was interesting to find that ECG-pBNP had better predictive abilities for mortality risk than ECG-BNP. Considering the high correlation between ECG-pBNP and ECG-BNP, the application of ECG-pBNP may be more feasible in future clinical practices.

Certain limitations should be mentioned. First, this was a retrospective study. A prospective study is needed to validate its efficacy in the community. Second, we did not evaluate echocardiography in these patients. The cause of elevated BNP/pBNP could be cardiac or noncardiac. Echocardiography could help to identify patients with systolic left ventricular dysfunction. Thirdly, it's important to note that ECG traits can exhibit variability across various racial backgrounds [52]. A comprehensive global study encompassing ethnically diverse populations is essential to authenticate the precision of ECG-BNP/pBNP. Lastly, there's a need for enhanced transparency in our ECG-BNP/pBNP methodology, given the enigmatic nature of the Deep Learning Model's "black box" [53]. Further research endeavors should delve into the correlation and comprehensibility of ECG characteristics concerning BNP/pBNP.

In conclusion, the assessment of BNP/pBNP using the AI-enabled ECG is a rapid and inexpensive method. This method can allow us to identify patients with high BNP/pBNP and may be considered an alternative option for plasma BNP/pBNP. ECG-pBNP provides a prognostic value in long-term mortality.

## Figures and Tables

**Figure 1 diagnostics-13-02723-f001:**
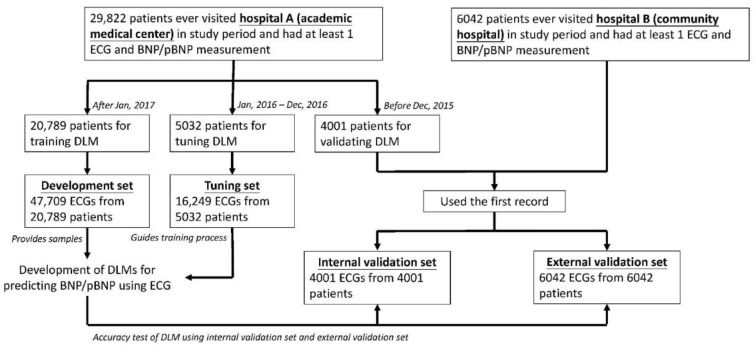
**Development, tuning, internal validation, and external validation set generation and ECG labeling of BNP/pBNP.** Schematic of the dataset creation and analysis strategy, which was devised to assure a robust and reliable dataset for the training, validating, and testing of the network. Once a patient’s data were placed in one of the datasets, that individual’s data were used only in that set, avoiding ‘cross-contamination’ among the training, validation, and test datasets. The details of the flow chart and how each of the datasets were used are described in the Methods.

**Figure 2 diagnostics-13-02723-f002:**
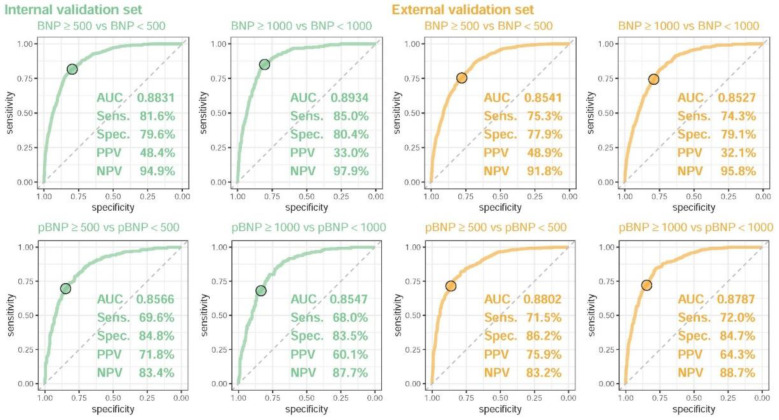
**The ROC curve of DLM predictions based on ECG to detect mild and severe abnormal BNP/pBNP.** Mild and severe abnormal BNP/pBNP were defined as actual BNP/pBNP of ≥500 and ≥1000, respectively. The operating point was selected based on the maximum Youden’s index in the tuning set and are presented using a circle mark; it was then used to calculate the area under the ROC curve (AUC), sensitivity (Sens.), specificity (Spec.), positive predictive value (PPV), and negative predictive value (NPV).

**Figure 3 diagnostics-13-02723-f003:**
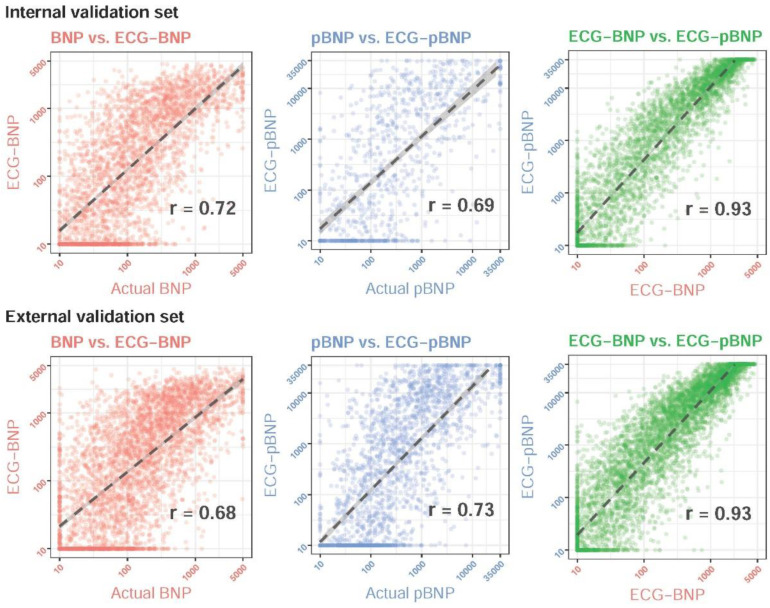
**Scatter plots of predicted BNP/pBNP (ECG-BNP/ECG-pBNP) and actual BNP/pBNP.** The *x*-axis and the *y*-axis are presented on a log scale. Red and blue colors represent BNP and pBNP, respectively. We presented the Pearson correlation coefficients (r) on a log scale to demonstrate the accuracy of the DLMs. The black lines with 95% conference intervals were fitted via simple linear regression on the log scale.

**Figure 4 diagnostics-13-02723-f004:**
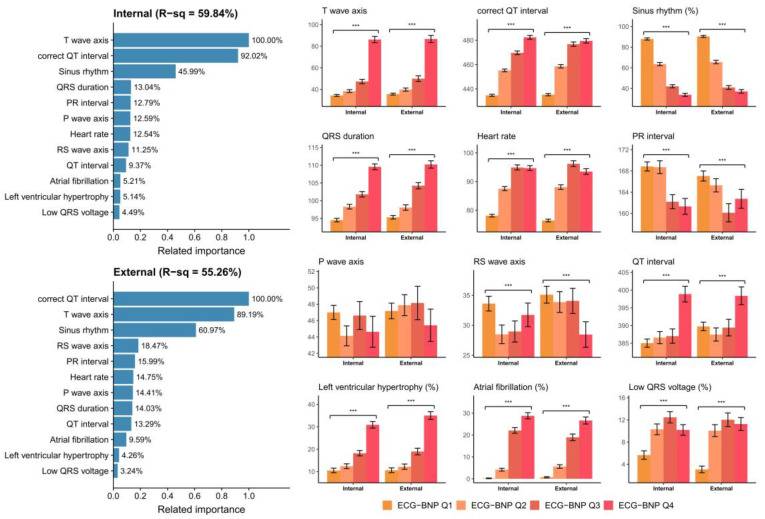
**Relationship between the most important ECG features and predicted BNP (ECG-BNP).** The related importance is based on the information gain of the XGB model, and the R-square (R-sq) is the coefficient of determination to use selected ECG features for predicting ECG-BNP on a log scale. The analyses were conducted in both the internal and external validation sets. (***: *p* for trend < 0.001).

**Figure 5 diagnostics-13-02723-f005:**
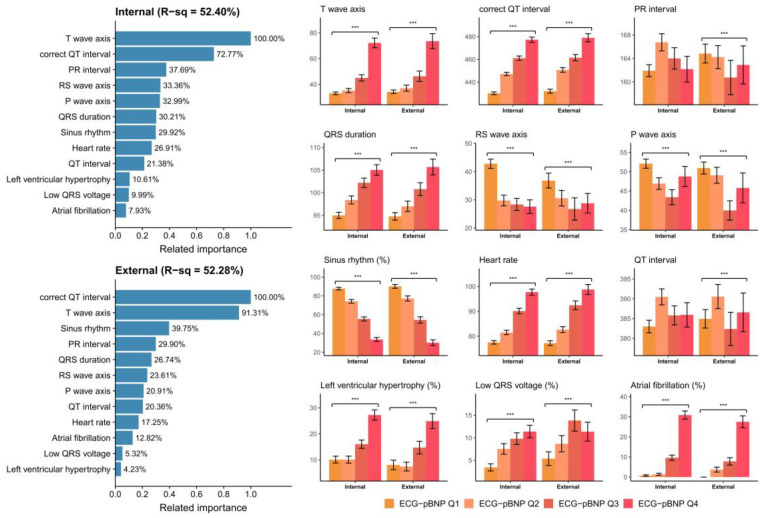
**Relationship between the most important ECG features and predicted pBNP (ECG-pBNP).** The related importance is based on the information gain of the XGB model, and the R-square (R-sq) is the coefficient of determination to use selected ECG features for predicting ECG-pBNP on a log scale. The analyses were conducted in both the internal and external validation sets. (***: *p* for trend < 0.001).

**Figure 6 diagnostics-13-02723-f006:**
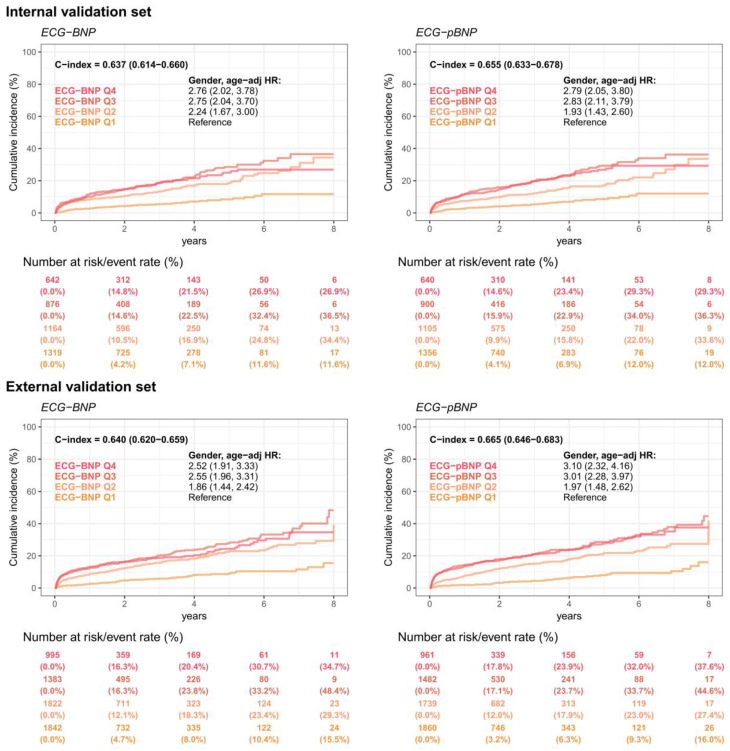
**Long-term incidence of developing mortality events stratified by ECG-BNP or ECG-pBNP.** The analyses were conducted in both the internal and external validation sets. The table shows the at-risk population and cumulative risk for the given time intervals in each risk stratification.

**Table 1 diagnostics-13-02723-t001:** Baseline characteristics in the internal and external validation sets.

	Internal Validation Set (*n* = 4001)		External Validation Set (*n* = 6042)	
	BNP Subset (*n* = 3090)	pBNP Subset (*n* = 911)	BNP Subset (*n* = 3966)	pBNP Subset (*n* = 2076)
**BNP/pBNP profile**				
mean ± SD in pg/mL	393.4 ± 789.5	2523.5 ± 6836.3	431.0 ± 798.2	2345.6 ± 6444.2
<500 pg/mL	2503 (81.0%)	585 (64.2%)	3095 (78.0%)	1291 (62.2%)
500–999 pg/mL	273 (8.8%)	82 (9.0%)	404 (10.2%)	209 (10.1%)
≥1000 pg/mL	314 (10.2%)	244 (26.8%)	467 (11.8%)	576 (27.7%)
**Demographics**				
Sex (male)	1639 (53.0%)	452 (49.6%)	2078 (52.4%)	1069 (51.5%)
Age (years)	69.3 ± 15.3	68.9 ± 15.5	74.0 ± 15.9	68.9 ± 18.3
BMI (kg/m^2^)	24.5 ± 4.4	24.7 ± 4.3	24.2 ± 4.4	24.2 ± 4.3
**Disease history**				
DM	1186 (38.4%)	386 (42.4%)	1578 (39.8%)	702 (33.8%)
HTN	1892 (61.2%)	609 (66.8%)	2722 (68.6%)	1173 (56.5%)
HLP	1310 (42.4%)	394 (43.2%)	1782 (44.9%)	716 (34.5%)
CKD	1345 (43.5%)	590 (64.8%)	1833 (46.2%)	1064 (51.3%)
AMI	191 (6.2%)	36 (4.0%)	160 (4.0%)	95 (4.6%)
STK	724 (23.4%)	236 (25.9%)	1118 (28.2%)	429 (20.7%)
CAD	1235 (40.0%)	397 (43.6%)	1518 (38.3%)	679 (32.7%)
HF	708 (22.9%)	134 (14.7%)	1036 (26.1%)	289 (13.9%)
Afib	380 (12.3%)	94 (10.3%)	526 (13.3%)	180 (8.7%)
COPD	827 (26.8%)	252 (27.7%)	1410 (35.6%)	495 (23.8%)

Abbreviations: SD, standard deviation; BNP, B-type natriuretic peptide; pBNP: N-terminal pro-brain natriuretic peptide; BMI: body mass index; DM: diabetes mellitus; HTN: hypertension; HLP: hyperlipidemia; CKD: chronic kidney disease; AMI: acute myocardial infarction; STK: stroke, CAD: coronary artery disease; HF: heart failure; Afib: atrial fibrillation; COPD: chronic obstructive pulmonary disease.

## Data Availability

Not applicable.

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
