# Peer review of "Artificial Intelligence-Enabled Electrocardiography Detects B-Type Natriuretic Peptide and N-Terminal Pro-Brain Natriuretic Peptide"

_diagnostics, 2023, doi:10.3390/diagnostics13172723_

Round 1
Reviewer 1 Report
The article is well written and the conclusions are clear and sound. But there are three major concerns.
1- The related work section specially those that used DL for this type of diagnosis or even diagnosis using ECG signals is missing by a great extent. Work on this to show the necessity of the research.
2- The proposed method is very briefly explained and it is even hard to find the used architecture. Explain it in more detail. preferably with a drawing. And explain why the architecture is used. The same for the other parts of the solution including XGB.
3- The main contribution of the paper will be shown when compared with previous approach and findings. Please include a comparison with previous works and findings section.
Author Response
Reviewer 1
The article is well written and the conclusions are clear and sound. But there are three major concerns.
1- The related work section specially those that used DL for this type of diagnosis or even diagnosis using ECG signals is missing by a great extent. Work on this to show the necessity of the research.
Reply: Thanks for your comments.
Thank you for the reviewer's suggestions. In order to enhance the readers' comprehension of the content, we have included detailed information about the deep learning model and the training process in the manuscript. This portion is written in lines 173-222.
Supplementary Figure 1 illustrates the architecture of our Deep Learning Model (DLM). Each electrocardiogram (ECG) was captured in the standard 12-lead format, comprising sequences of 5000 data points. These sequences were used to create a matrix of size 5000 × 12. The input format of this architecture was a 4096 × 12 matrix. During the training process, we randomly selected sequences of length 4096 as input. In the inference stage, two overlapping sequences of 4096 were utilized from both the beginning and the end to generate predictions, which were then averaged to produce the final prediction.
We defined a "residual module" as a neural combination with a constant factor k. This module included the following components: (1) a 1 × 1 convolutional layer with k/4 filters for dimension reduction, (2) a batch normalization layer for normalization, (3) a rectified linear unit (ReLU) layer for introducing non-linearity, (4) a 3 × 1 convolutional layer with k/4 filters to extract features, (5) a batch normalization layer for normalization, (6) a ReLU layer for non-linearity, (7) a 3 × 1 convolutional layer with 4K filters to further extract features, (8) a 1 × 1 convolutional layer with k filters to restore feature shape, (9) a batch normalization layer for normalization, (10) a ReLU layer for non-linearity, and (11) a squeeze-and-excitation (SE) module for feature weighting. The SE module comprised: (1) an average global pooling layer, (2) a fully-connected layer with k/r neurons, and (3) an-other fully-connected layer with k neurons. The constant r remained fixed at 8 across all experiments. The residual module concluded with a shortcut connection, creating direct connections between each layer and all subsequent layers.
When there's a change in the size of feature maps, concatenating the residual module becomes infeasible. As a solution, our architecture employs a "pool module" to facilitate the concatenation of each residual module during down-sampling. This module consists of similar concatenated layers as the residual modules, but the stride of the 3 × 1 convolution layer is altered to 2 × 1. Down-sampling is achieved through an average pooling layer with a 2 × 1 kernel size and stride. These components are integrated using the concatenation function.
The input data first pass through a batch normalization layer, followed by an 11 × 1 convolution layer with a 2 × 1 stride and 16 filters, another batch normalization layer, a ReLU layer, and a pool module. Subsequently, the data traverse a series of residual modules and pool modules, yielding a 32 × 12 × 1024 array. This is followed by a global pool-ing layer and the last residual module. The array is then split into 12 lead-specific feature maps, each containing 1024 features. These feature maps undergo processing via a fully-connected layer with 1 neuron to generate lead-specific predictions.
To enhance the interpretive capacity of the DLM, an attention mechanism based on a hierarchical attention network is employed to combine these blocks. The attention module involves a fully connected layer with 8 neurons, followed by a batch normalization layer, a ReLU layer, and another fully-connected layer with 1 neuron that generates weights for each lead. Attention scores are computed for each ECG lead, and then integrated and standardized through the final linear output layer. These standardized attention scores are used to weight the outputs of the 12 ECG leads via simple multiplication. The weighted outputs are summed and processed through a prediction module to yield the final prediction value.
During training, these DLMs use a batch size of 32 and an initial learning rate of 0.001, with an Adam optimizer employing standard parameters (β1 = 0.9 and β2 = 0.999). The learning rate undergoes decay by a factor of 10 whenever the validation cohort's loss plateaus after an epoch. To curb overfitting, we employ early stopping by saving the net-work after each epoch and selecting the DLMs with the lowest loss on the validation co-hort. In this study, the sole regularization method for preventing overfitting is L2 regularization with a coefficient of 10-4.
Supplementary Figure 1|The implementation of our deep learning model. The model architectures of the deep learning model for analyzing ECG.
2- The proposed method is very briefly explained and it is even hard to find the used architecture. Explain it in more detail. preferably with a drawing. And explain why the architecture is used. The same for the other parts of the solution including XGB.
Reply: Thanks for your comments.
Thank you for the reviewer's suggestions. To enhance the readers' understanding of the manuscript, we have supplemented detailed information about the deep learning model in the manuscript. Additionally, we have visualized this information in Supplementary Figure 1. This portion is written in lines 136-137, 173-222 and 241-243.
lines 136-137
These features were used to train an extreme gradient boosting (XGB) model, and the DLM was trained via raw ECG traces.
lines 173-222
Supplementary Figure 1 illustrates the architecture of our Deep Learning Model (DLM). Each electrocardiogram (ECG) was captured in the standard 12-lead format, comprising sequences of 5000 data points. These sequences were used to create a matrix of size 5000 × 12. The input format of this architecture was a 4096 × 12 matrix. During the training process, we randomly selected sequences of length 4096 as input. In the inference stage, two overlapping sequences of 4096 were utilized from both the beginning and the end to generate predictions, which were then averaged to produce the final prediction.
We defined a "residual module" as a neural combination with a constant factor k. This module included the following components: (1) a 1 × 1 convolutional layer with k/4 filters for dimension reduction, (2) a batch normalization layer for normalization, (3) a rectified linear unit (ReLU) layer for introducing non-linearity, (4) a 3 × 1 convolutional layer with k/4 filters to extract features, (5) a batch normalization layer for normalization, (6) a ReLU layer for non-linearity, (7) a 3 × 1 convolutional layer with 4K filters to further extract features, (8) a 1 × 1 convolutional layer with k filters to restore feature shape, (9) a batch normalization layer for normalization, (10) a ReLU layer for non-linearity, and (11) a squeeze-and-excitation (SE) module for feature weighting. The SE module comprised: (1) an average global pooling layer, (2) a fully-connected layer with k/r neurons, and (3) an-other fully-connected layer with k neurons. The constant r remained fixed at 8 across all experiments. The residual module concluded with a shortcut connection, creating direct connections between each layer and all subsequent layers.
When there's a change in the size of feature maps, concatenating the residual module becomes infeasible. As a solution, our architecture employs a "pool module" to facilitate the concatenation of each residual module during down-sampling. This module consists of similar concatenated layers as the residual modules, but the stride of the 3 × 1 convolution layer is altered to 2 × 1. Down-sampling is achieved through an average pooling layer with a 2 × 1 kernel size and stride. These components are integrated using the concatenation function.
The input data first pass through a batch normalization layer, followed by an 11 × 1 convolution layer with a 2 × 1 stride and 16 filters, another batch normalization layer, a ReLU layer, and a pool module. Subsequently, the data traverse a series of residual modules and pool modules, yielding a 32 × 12 × 1024 array. This is followed by a global pool-ing layer and the last residual module. The array is then split into 12 lead-specific feature maps, each containing 1024 features. These feature maps undergo processing via a fully-connected layer with 1 neuron to generate lead-specific predictions.
To enhance the interpretive capacity of the DLM, an attention mechanism based on a hierarchical attention network is employed to combine these blocks. The attention module involves a fully connected layer with 8 neurons, followed by a batch normalization layer, a ReLU layer, and another fully-connected layer with 1 neuron that generates weights for each lead. Attention scores are computed for each ECG lead, and then integrated and standardized through the final linear output layer. These standardized attention scores are used to weight the outputs of the 12 ECG leads via simple multiplication. The weighted outputs are summed and processed through a prediction module to yield the final prediction value.
During training, these DLMs use a batch size of 32 and an initial learning rate of 0.001, with an Adam optimizer employing standard parameters (β1 = 0.9 and β2 = 0.999). The learning rate undergoes decay by a factor of 10 whenever the validation cohort's loss plateaus after an epoch. To curb overfitting, we employ early stopping by saving the net-work after each epoch and selecting the DLMs with the lowest loss on the validation co-hort. In this study, the sole regularization method for preventing overfitting is L2 regularization with a coefficient of 10-4.
Supplementary Figure 1|The implementation of our deep learning model. The model architectures of the deep learning model for analyzing ECG.
lines 241-243
The importance analysis of ECG features was initiated with an eXtreme gradient boosting (XGB) model for predicting ECG-BNP/ECG-pBNP, resulting in the presentation of feature importance rankings.
3- The main contribution of the paper will be shown when compared with previous approach and findings. Please include a comparison with previous works and findings section.
Reply: Thanks for your comments.
Thank you for the reviewer's suggestions. We had previously conducted a search to determine if there were any similar studies. However, currently, there appears to be no other published research by other researchers on predicting BNP/pBNP using electrocardiograms. As a result, a direct comparison is not feasible at this time.

Reviewer 2 Report
1- The authors do not show the different patterns of the ECG.
2- No details about the deep learning model.
3- Statistical analysis is not enough and needs to be explained.
4- No mention of how the DL model was evaluated.
5- No confusion matrix for the evaluation of the model.
6- No explanation for the selected values of BNP values.
7- Details of the used deep learning model layers should be added.
8- Are there any confounding factors in the anthropometric data? Please explain.
9-
Minor corrections are required.
Author Response
Reviewer 2
1- The authors do not show the different patterns of the ECG.
Reply: Thanks for your comments.
In order to make it clearer to the readers which 31 diagnostic patterns are being referred to, we will add them to the Data Collection section in line 138-146.
line 138-146
Pattern classes included abnormal T wave, atrial fibrillation, atrial flutter, atrial premature complex, complete AV block, complete left bundle branch block, complete right bundle branch block, first degree AV block, incomplete left bundle branch block, incomplete right bundle branch block, ischemia/infarction, junctional rhythm, left anterior fascicular block, left atrial enlargement, left axis deviation, left posterior fascicular block, left ventricular hypertrophy, low QRS voltage, pacemaker rhythm, prolonged QT interval, right atrial enlargement, right ventricular hypertrophy, second degree AV block, sinus bradycardia, sinus pause, sinus rhythm, sinus tachycardia, supraventricular tachycardia, ventricular premature complex, ventricular tachycardia, and Wolff–Parkinson–White syndrome.
2- No details about the deep learning model.
Reply: Thanks for your comments.
Thank you for the reviewer's suggestions. In order to enhance the readers' comprehension of the content, we have included detailed information about the deep learning model and the training process in the manuscript. This portion is written in lines 173-222.
lines 173-222
Supplementary Figure 1 illustrates the architecture of our Deep Learning Model (DLM). Each electrocardiogram (ECG) was captured in the standard 12-lead format, comprising sequences of 5000 data points. These sequences were used to create a matrix of size 5000 × 12. The input format of this architecture was a 4096 × 12 matrix. During the training process, we randomly selected sequences of length 4096 as input. In the inference stage, two overlapping sequences of 4096 were utilized from both the beginning and the end to generate predictions, which were then averaged to produce the final prediction.
We defined a "residual module" as a neural combination with a constant factor k. This module included the following components: (1) a 1 × 1 convolutional layer with k/4 filters for dimension reduction, (2) a batch normalization layer for normalization, (3) a rectified linear unit (ReLU) layer for introducing non-linearity, (4) a 3 × 1 convolutional layer with k/4 filters to extract features, (5) a batch normalization layer for normalization, (6) a ReLU layer for non-linearity, (7) a 3 × 1 convolutional layer with 4K filters to further extract features, (8) a 1 × 1 convolutional layer with k filters to restore feature shape, (9) a batch normalization layer for normalization, (10) a ReLU layer for non-linearity, and (11) a squeeze-and-excitation (SE) module for feature weighting. The SE module comprised: (1) an average global pooling layer, (2) a fully-connected layer with k/r neurons, and (3) an-other fully-connected layer with k neurons. The constant r remained fixed at 8 across all experiments. The residual module concluded with a shortcut connection, creating direct connections between each layer and all subsequent layers.
When there's a change in the size of feature maps, concatenating the residual module becomes infeasible. As a solution, our architecture employs a "pool module" to facilitate the concatenation of each residual module during down-sampling. This module consists of similar concatenated layers as the residual modules, but the stride of the 3 × 1 convolution layer is altered to 2 × 1. Down-sampling is achieved through an average pooling layer with a 2 × 1 kernel size and stride. These components are integrated using the concatenation function.
The input data first pass through a batch normalization layer, followed by an 11 × 1 convolution layer with a 2 × 1 stride and 16 filters, another batch normalization layer, a ReLU layer, and a pool module. Subsequently, the data traverse a series of residual modules and pool modules, yielding a 32 × 12 × 1024 array. This is followed by a global pool-ing layer and the last residual module. The array is then split into 12 lead-specific feature maps, each containing 1024 features. These feature maps undergo processing via a fully-connected layer with 1 neuron to generate lead-specific predictions.
To enhance the interpretive capacity of the DLM, an attention mechanism based on a hierarchical attention network is employed to combine these blocks. The attention module involves a fully connected layer with 8 neurons, followed by a batch normalization layer, a ReLU layer, and another fully-connected layer with 1 neuron that generates weights for each lead. Attention scores are computed for each ECG lead, and then integrated and standardized through the final linear output layer. These standardized attention scores are used to weight the outputs of the 12 ECG leads via simple multiplication. The weighted outputs are summed and processed through a prediction module to yield the final prediction value.
During training, these DLMs use a batch size of 32 and an initial learning rate of 0.001, with an Adam optimizer employing standard parameters (β1 = 0.9 and β2 = 0.999). The learning rate undergoes decay by a factor of 10 whenever the validation cohort's loss plateaus after an epoch. To curb overfitting, we employ early stopping by saving the net-work after each epoch and selecting the DLMs with the lowest loss on the validation co-hort. In this study, the sole regularization method for preventing overfitting is L2 regularization with a coefficient of 10-4.
Supplementary Figure 1|The implementation of our deep learning model. The model architectures of the deep learning model for analyzing ECG.
3- Statistical analysis is not enough and needs to be explained.
Reply: Thanks for your comments.
Thank you for the reviewer's suggestions. We have revised the presentation of the statistical analysis to enhance the readers' understanding. This portion is written in lines 230-249.
The presentation of patient characteristics included means and standard deviations for continuous variables, along with numbers and percentages for categorical variables. The statistical analysis was conducted using R version 3.4.4, and the implementation of our Deep Learning Models (DLMs) was done using the MXNet package version 1.3.0. To evaluate the performance of the DLMs in identifying mild and severe abnormal BNP/pBNP, we employed receiver operating characteristic (ROC) curves and calculated areas under the curve (AUCs). Sensitivity, specificity, positive predictive value, and negative predictive value were determined based on the maximum Youden's index from the tuning set. The selected operating point was consistently used for analysis in both internal and external validation sets. For continuous analysis, scatter plots and Pearson correlation coefficients (r) were used to compare the predictions from the electrocardiogram (ECG) with actual BNP/pBNP values. The importance analysis of ECG features was initiated with an eXtreme gradient boosting (XGB) model for predicting ECG-BNP/ECG-pBNP, resulting in the presentation of feature importance rankings. The relationship between ECG features and ECG-BNP/ECG-pBNP was visually represented using a bar chart with quartiles. In the context of mortality analysis, a Kaplan‒Meier (KM) curve was generated to as-sess the prognostic contribution of ECG-BNP/ECG-pBNP. Additionally, Cox proportional hazards models were fitted to calculate grouping hazard ratios (HRs) along with corresponding 95% confidence intervals (95% CI). The C-index was employed as a global indicator to compare the predictive performance between ECG-BNP and ECG-pBNP.
4- No mention of how the DL model was evaluated.
Reply: Thanks for your comments.
Thank you for the reviewer's suggestions. We have now included the method for evaluating the accuracy of the deep learning model in the "Statistical Analysis" section in line 233-238.
To evaluate the performance of the DLMs in identifying mild and severe abnormal BNP/pBNP, we employed receiver operating characteristic (ROC) curves and calculated areas under the curve (AUCs). Sensitivity, specificity, positive predictive value, and negative predictive value were determined based on the maximum Youden's index from the tuning set.
5- No confusion matrix for the evaluation of the model.
Reply: Thanks for your comments.
Thank you for the reviewer's suggestions. The reason we did not present a confusion matrix is that we utilized the ROC curve in Figure 2 as our method for evaluating model performance. Sensitivity, specificity, positive predictive value, and negative predictive value are also depicted, and these metrics are derived from the confusion matrix. Hence, in order to avoid redundant information, we opted to utilize Figure 2 for representation. Thank you.
6- No explanation for the selected values of BNP values.
BNP (B-type natriuretic peptide) and pBNP (pro-B-type natriuretic peptide) are biomarkers associated with cardiac function and cardiovascular diseases. These biomarkers are commonly used to assess heart health and the risk of cardiovascular conditions, particularly heart failure. They play a significant role in the diagnosis, treatment, and monitoring of these diseases. The biological significance of BNP and pBNP, as well as their associations with cardiac function and cardiovascular diseases, are described in the background. How these biomarkers are generated in physiological and pathological conditions, and their correlation with heart health status, are all elaborated upon in the background. Since there are no universal cutoff points for BNP and pBNP, we divided the data into three categories based on BNP/pBNP by the same values: normal BNP/pBNP (<500 pg/mL), mild abnormal BNP/pBNP (500–999 pg/mL), and severe abnormal BNP/pBNP (≥1000 pg/mL). This portion is written in lines 147-151.
lines 147-151
Blood BNP and pBNP were based on central laboratory methods. Since there are no universal cutoff points for BNP and pBNP, we divided the data into three categories based on BNP/pBNP by the same values: normal BNP/pBNP (<500 pg/mL), mild abnormal BNP/pBNP (500–999 pg/mL), and severe abnormal BNP/pBNP (≥1000 pg/mL).
7- Details of the used deep learning model layers should be added.
Thank you for the reviewer's suggestions. In order to enhance the readers' comprehension of the content, we have included detailed information about the deep learning model and the training process in the manuscript. This portion is written in lines 173-222.
lines 173-222
Supplementary Figure 1 illustrates the architecture of our Deep Learning Model (DLM). Each electrocardiogram (ECG) was captured in the standard 12-lead format, comprising sequences of 5000 data points. These sequences were used to create a matrix of size 5000 × 12. The input format of this architecture was a 4096 × 12 matrix. During the training process, we randomly selected sequences of length 4096 as input. In the inference stage, two overlapping sequences of 4096 were utilized from both the beginning and the end to generate predictions, which were then averaged to produce the final prediction.
We defined a "residual module" as a neural combination with a constant factor k. This module included the following components: (1) a 1 × 1 convolutional layer with k/4 filters for dimension reduction, (2) a batch normalization layer for normalization, (3) a rectified linear unit (ReLU) layer for introducing non-linearity, (4) a 3 × 1 convolutional layer with k/4 filters to extract features, (5) a batch normalization layer for normalization, (6) a ReLU layer for non-linearity, (7) a 3 × 1 convolutional layer with 4K filters to further extract features, (8) a 1 × 1 convolutional layer with k filters to restore feature shape, (9) a batch normalization layer for normalization, (10) a ReLU layer for non-linearity, and (11) a squeeze-and-excitation (SE) module for feature weighting. The SE module comprised: (1) an average global pooling layer, (2) a fully-connected layer with k/r neurons, and (3) an-other fully-connected layer with k neurons. The constant r remained fixed at 8 across all experiments. The residual module concluded with a shortcut connection, creating direct connections between each layer and all subsequent layers.
When there's a change in the size of feature maps, concatenating the residual module becomes infeasible. As a solution, our architecture employs a "pool module" to facilitate the concatenation of each residual module during down-sampling. This module consists of similar concatenated layers as the residual modules, but the stride of the 3 × 1 convolution layer is altered to 2 × 1. Down-sampling is achieved through an average pooling layer with a 2 × 1 kernel size and stride. These components are integrated using the concatenation function.
The input data first pass through a batch normalization layer, followed by an 11 × 1 convolution layer with a 2 × 1 stride and 16 filters, another batch normalization layer, a ReLU layer, and a pool module. Subsequently, the data traverse a series of residual modules and pool modules, yielding a 32 × 12 × 1024 array. This is followed by a global pool-ing layer and the last residual module. The array is then split into 12 lead-specific feature maps, each containing 1024 features. These feature maps undergo processing via a fully-connected layer with 1 neuron to generate lead-specific predictions.
To enhance the interpretive capacity of the DLM, an attention mechanism based on a hierarchical attention network is employed to combine these blocks. The attention module involves a fully connected layer with 8 neurons, followed by a batch normalization layer, a ReLU layer, and another fully-connected layer with 1 neuron that generates weights for each lead. Attention scores are computed for each ECG lead, and then integrated and standardized through the final linear output layer. These standardized attention scores are used to weight the outputs of the 12 ECG leads via simple multiplication. The weighted outputs are summed and processed through a prediction module to yield the final prediction value.
During training, these DLMs use a batch size of 32 and an initial learning rate of 0.001, with an Adam optimizer employing standard parameters (β1 = 0.9 and β2 = 0.999). The learning rate undergoes decay by a factor of 10 whenever the validation cohort's loss plateaus after an epoch. To curb overfitting, we employ early stopping by saving the net-work after each epoch and selecting the DLMs with the lowest loss on the validation co-hort. In this study, the sole regularization method for preventing overfitting is L2 regularization with a coefficient of 10-4.
Supplementary Figure 1|The implementation of our deep learning model. The model architectures of the deep learning model for analyzing ECG.
8- Are there any confounding factors in the anthropometric data? Please explain.
Reply: Thanks for your comments.
Regarding anthropometric data, we believe it can encompass the following aspects:
(1) Data on human body size and shape serve as the foundation for constructing all digital human models. Measurements such as height, weight, etc., are obtained through instruments.
(2) Blood BNP and pBNP measurements were conducted using central laboratory methods. In theory, errors are unlikely unless there is contamination of the blood samples.
(3) The 12-lead electrocardiogram tests were performed by trained operators, ensuring accurate measurements and minimizing the potential for issues.
We have carefully considered these measurement aspects to ensure the accuracy and reliability of our data. During the analysis, we took into account potential confounding factors to better reflect real-world situations in our study results.

Reviewer 3 Report
Diagnostics
Recommendation Acceptance: Accept
Comment:
This manuscript introduces the AI-enabled ECG system for predicting cardiovascular morbidity and mortality. The important thing in training using deep learning models (DLMs) is learning through various information, and the author has proven this using a large number of patient data. Various condition experiments were performed to demonstrate the reliability of AI-enabled ECG. This could identify patients with high BNP/pBNP, suggesting a new plasma BNP/pBNP method.
As a result, we recommend this manuscript be accepted in Diagnostics.
Author Response
Thanks for your comments.
Round 2
Reviewer 1 Report
I am satisfied by the responses